# Transient *Ascaris suum* larval migration induces intractable chronic pulmonary disease and anemia in mice

Yifan Wu[1,2], Evan Li[2], Morgan Knight[2], Grace Adeniyi-Ipadeola[1], Li-zhen Song[2], Alan R. Burns[3], Ana Clara Gazzinelli-Guimaraes[4], Ricardo Fujiwara[4], Maria Elena Bottazzi[1,5,6], Jill E. Weatherhead[1,5,7] *

1 Department of Pediatrics, Pediatric Tropical Medicine, Baylor College of Medicine, Houston, Texas, United States of America, 2 Department of Medicine, Pathology and Immunology, and the Biology of Inflammation Center, Baylor College of Medicine, Houston, Texas, United States of America, 3 College of Optometry, University of Houston, Houston, Texas, United States of America, 4 Departamento de Parasitologia, Universidade Federal de Minas Gerais, Belo Horizonte, Brazil, 5 National School of Tropical Medicine, Baylor College of Medicine, Houston, Texas, United States of America, 6 Texas Children's Hospital Center for Vaccine Development, Baylor College of Medicine, Houston, Texas, United States of America, 7 Department of Medicine, Infectious Diseases, Baylor College of Medicine, Houston, Texas, United States of America

* weatherh@bcm.edu

**Data Availability Statement:** All relevant data are within the manuscript and its Supporting Information files.

## Abstract

Ascariasis is one of the most common infections in the world and associated with significant global morbidity. *Ascaris* larval migration through the host's lungs is essential for larval development but leads to an exaggerated type-2 host immune response manifesting clinically as acute allergic airway disease. However, whether *Ascaris* larval migration can subsequently lead to chronic lung diseases remains unknown. Here, we demonstrate that a single episode of *Ascaris* larval migration through the host lungs induces a chronic pulmonary syndrome of type-2 inflammatory pathology and emphysema accompanied by pulmonary hemorrhage and chronic anemia in a mouse model. Our results reveal that a single episode of *Ascaris* larval migration through the host lungs leads to permanent lung damage with systemic effects. Remote episodes of ascariasis may drive non-communicable lung diseases such as asthma, chronic obstructive pulmonary disease (COPD), and chronic anemia in parasite endemic regions.

## Author summary

Ascariasis is the most common helminth infection and leads to significant global morbidity. Transient *Ascaris* larval migration through the host's lungs is essential for larval development but leads to an exaggerated type-2 host immune response. Our work demonstrates that transient *Ascaris* spp. larval migration through the lungs has significant long-term consequences including changes in lung structure and function as well as vascular damage causing chronic lung disease and anemia. We propose that *Ascaris* spp. larval migration through the host lungs is a risk factor for the development of chronic lung disease and anemia in parasite-endemic regions globally.

**Funding:** This project was funded by NIH K08 AI143968-01 (to JW) as well as the serial block-face imaging core NIH/NEI P30 EY007551 (to AB). This project also received funding from the Mouse Metabolism and Phenotyping Core at Baylor College of Medicine which receives support from NIH (UM1HG006348, R01DK114356, R01HL130249). The Mouse Metabolism and Phenotyping Core receives additional funding from Baylor College of Medicine as a member of its Advanced Technology Cores. This project was further supported by the Cytometry and Cell Sorting Core at Baylor College of Medicine with funding from the CPRIT Core Facility Support Award (CPRIT-RP180672), the NIH (CA125123 and RR024574) and the assistance of Joel M. Sederstrom. The funders had no role in study design, data collection and analysis, decision to publish, or preparation of the manuscript.

**Competing interests:** The authors have declared that no competing interests exist.

## Introduction

Ascariasis is the most common helminth infection globally, affecting approximately 500 million people living in low- and middle-income countries (LMIC) and is associated with significant global morbidity, equating to nearly 800,000 disability-adjusted life years (DALYs)[1–3]. In endemic regions, children are often infected in infancy and endure recurrent infection throughout childhood[4,5]. Children become infected with either *Ascaris lumbricoides*, human roundworm, or *Ascaris suum*, porcine roundworm, via oral ingestion of eggs that are ubiquitous within the environment. Once ingested, *Ascaris* spp. larvae hatch in the intestines and migrate to the liver followed by the lungs via the systemic circulation. Once in the lungs, the larvae penetrate through the endovasculature into the lung parenchyma where the larvae mature. The mature larvae move through the alveolar epithelium, ascend the bronchotracheal tree and are then swallowed back into the intestines where they develop into adult worms[6,7]. Completion of the two month larval migration cycle is essential for the parasite to develop into adult worms in the intestines[8]. Like other helminths, the immunomodulatory properties of *Ascaris* adult worms permit long-term occupancy in the host intestinal lumen up to two years [9,10].

The short-term consequence of the *Ascaris* spp. larval migration cycle is the development of pulmonary type-2 inflammation and allergic airway disease. *Ascaris* larval migration through the lungs can induce profound type-2 inflammatory infiltration with high levels of type-2 cytokines, severe airway hyperreactivity and mucous production in a mouse model[11]. This type-2 inflammatory infiltrate aims to reduce larval development and parasite burden but leads to transient allergic inflammation[11,12]. However, the long-term consequences of *Ascaris* larval migration through the lungs are unknown. Using a mouse model of *Ascaris suum* larval migration, a zoonotic human pathogen morphologically and genetically similar to *Ascaris lumbricoides*[13], our work demonstrates that transient *Ascaris* spp. larval migration through the lungs has significant long-term consequences including changes in lung structure and function as well as vascular damage causing chronic lung disease and anemia. We propose that *Ascaris* spp. larval migration through the host lungs is a risk factor for the development of chronic lung disease and anemia in parasite-endemic regions globally. These findings have significant implications for informing public policy and shaping strategies to reduce global parasite-induced morbidity.

## Results

### *Ascaris* larval migration induces chronic pulmonary type-2 inflammatory pathology in mice

Our previous work showed that a single *Ascaris* infection causes profound pulmonary type-2 inflammation consistent with acute allergic airway disease in a mouse model[11]. To determine the potential chronic effects of a single, transient *Ascaris suum* infection on the host lungs, we first assessed the development of chronic type-2 inflammation in the lungs[14]. Balb/c mice were challenged with 2500 *A.suum* eggs, the standard inoculum for ascariasis in a murine model[15], via oral gavage to assess for key type-2 inflammatory features at different time points post infection (p.i.) (**Fig 1**). As previously described[11], at day 12 p.i., *A.suum* induced a potent airway hyperresponsiveness characterized by increased respiratory system resistance ($R_{RS}$) (**Fig 1A**). $R_{Rs}$ was several times higher and remained significantly elevated at 2 months p.i. in *Ascaris*-challenged compared to PBS-challenged, naïve mice. Likewise, *Ascaris*-challenged mice had substantially elevated cell counts in the bronchoalveolar lavage fluid

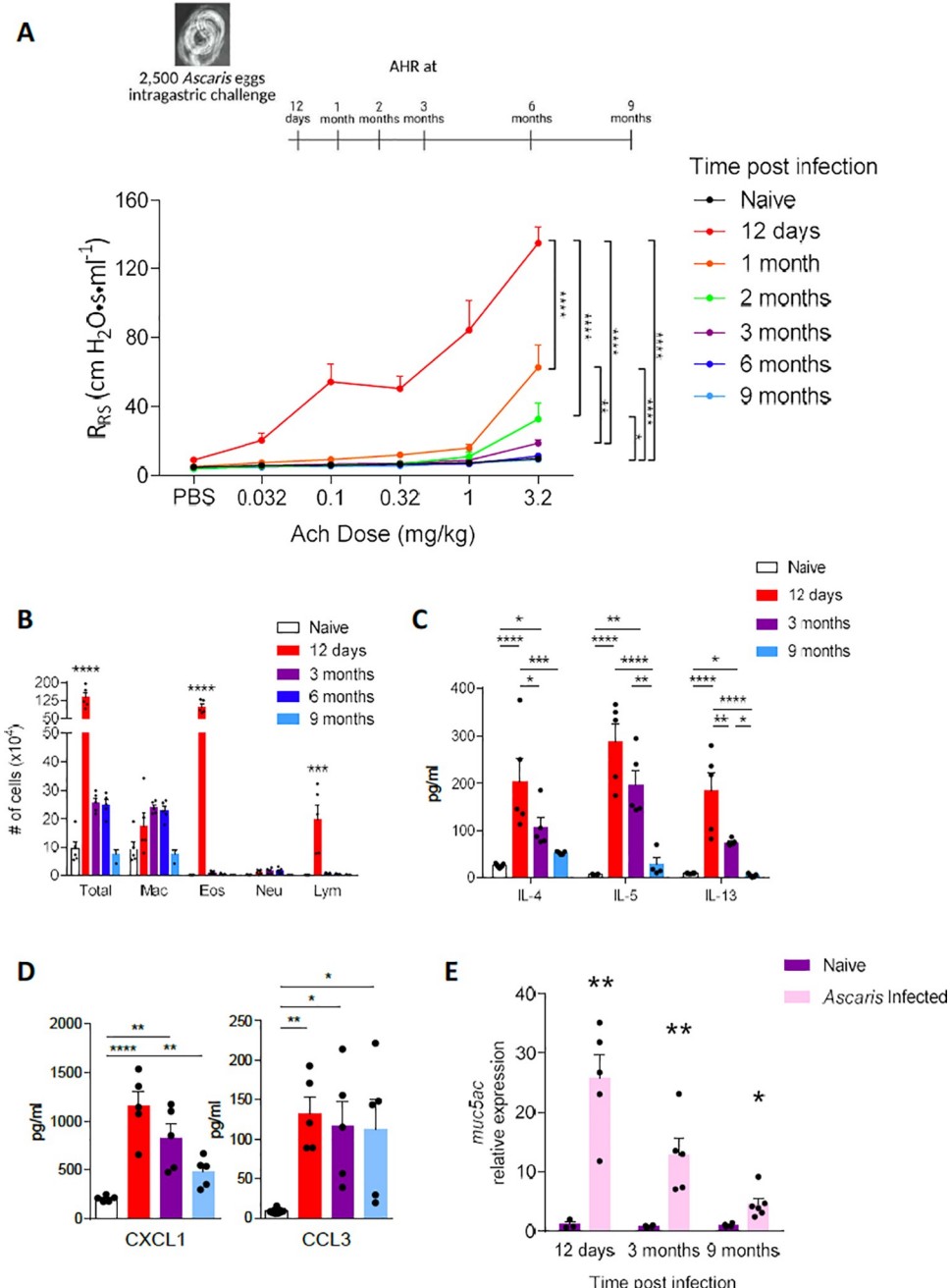

**Fig 1. *Ascaris* induces long term pulmonary type-2 inflammatory pathology in mice.** BALB/c mice were challenged by oral gavage with 2,500 eggs of *Ascaris* once or phosphate buffered saline (PBS), and type-2 inflammatory pathology was assessed for both *Ascaris*-infected and PBS naïve mice at standard intervals post-infection (p.i.) as indicated on the timeline. A representative PBS naïve mouse group is displayed for each experiment as the values at each time interval remained unchanged across naïve groups **(A)** Respiratory system resistance ($R_{RS}$) was assessed after intravenous injection of increasing doses of acetylcholine (Ach) to assesses airway hyperreactivity **(B)** Quantitation of cells from bronchoalveolar lavage fluid samples (mac: macrophages; eos: eosinophils; neu: neutrophils; lym: lymphocytes) to determine cellular composition over time **(C)** $T_H2$ cytokines including Interleukin (IL)-4, IL-5, IL-13 and **(D)** chemokines C-X-C motif chemokine ligand 1 (CXCL1) and C-C motif chemokine ligand 3 (CCL3) quantitated by Luminex and ELISA from deaggregated lung supernatants **(E)** mRNA level of mucin5AC (*muc*5AC) from lungs post infection at standard intervals p.i.. (n≥5, mean ± standard error of the mean (S.E.M), n.s.: not significant, *p<0.05, **p<0.01, ***p<0.001, ****p<0.0001, using one-way ANOVA followed by Tukey's test for multiple comparison. Data are representative of at least two independent experiments)

(BALF) at day 12 p.i., marked by increased eosinophils and lymphocytes (**Fig 1B**). The total BALF cell count remained elevated as far out as 6 months p.i. While the total eosinophil count decreased dramatically over time, macrophages remained elevated for at least 6 months.

Moreover, we detected increased production of type-2 cytokines (IL-4, IL-5 and IL-13) from de-aggregated lung that were elevated for at least 3 months p.i. and then sharply declined (**Fig 1C**), similar to the typical type-2 responses that result in allergic airway disease[14,16–19]. While it appears AHR and type-2 cytokines transition to baseline by 3 months p.i., other indicators of immune activation persisted beyond 3 months p.i. Chemokine levels including CXCL1, involved in leukocyte recruitment following helminth larval migration[20], and CCL3, associated with chronic macrophage recruitment during helminth infection and a marker of severe helminth disease[21,22], remained elevated as long as 9 months p.i. (**Fig 1D**). We also detected escalated expression of muc5ac for up to 9 months p.i. (**Fig 1E**), indicating on-going bronchial mucus production, a common finding in allergic airway disease[23]. The presence of goblet cell hyperplasia early in disease as previously described[11] as well as at 9 months post-infection using PAS staining (**S1 Fig**) corroborated the finding of sustained expression of muc5ac in *Ascaris* infected mice over time.

Remarkably, our results demonstrate that a single, self-limited *A.suum* infection, lasting no more than 14 days[15], induces a chronic pulmonary type-2 inflammatory phenotype with some features persisting for up to 9 months.

## *Ascaris* larval migration causes chronic structural and functional pulmonary changes consistent with chronic lung disease

In addition to the ongoing lung immune activation, we sought to determine the corresponding lung structural and functional changes. We discovered that after 12 days p.i., lungs from *Ascaris* infected mice were grossly larger than PBS-challenged, naïve mice (**Fig 2A**). We quantified post-expiratory aerated lung volume ($mm^3$) by microCT, which confirmed the increased lung volume at 12 d.p.i. that persisted up to 9 months p.i., despite normal lung remodeling associated with growth (**Fig 2B**). Histological evaluation of inflated *Ascaris* infected lungs was carried out to further characterize lung structure by assessing alveolar space morphology using mean linear intercept (MLI) quantification. Mice with previous *A.suum* infection developed increased MLI starting at 3 months p.i., which persisted up to 9 months p.i., demonstrating chronic alveolar wall destruction and enlargement of airspaces. Of note, while no *Ascaris* larvae were visualized in the lung tissue after 12 days p.i., the persistent dense inflammatory infiltrate around the bronchoalveolar bundles inhibited an accurate measurement of MLI prior to 3 months p.i. (**Fig 2C**). Low-magnification histopathology images at day 12 p.i. and 9 months p.i. demonstrate the changes in inflammatory infiltrate around the bronchoalveolar bundles over time from dense inflammatory infiltrate early in disease to the presence of alveolar macrophages late in the disease (**S2 Fig**). These results indicate that *Ascaris* spp. larvae induce irreversible alveolar damage consistent with emphysema, a type of chronic obstructive pulmonary disease (COPD) most often seen in the context of tobacco smoking[24–26].

An important physiological correlate of emphysema is enhanced lung compliance and dynamic airway collapse during exhalation that leads to air trapping, enhanced lung volume, and in affected persons, shortness of breath. To determine if similar functional changes occurred in our mice, lung compliance was assessed. *A.suum* infected mice were found to have increased lung compliance compared to PBS-challenged mice as assessed by total lung capacity at a standard lung pressure starting at 12 days and persisting up to 9 months p.i., despite normal age-related growth (**Fig 2D**)[27]. Together, our results demonstrate that transient *Ascaris*

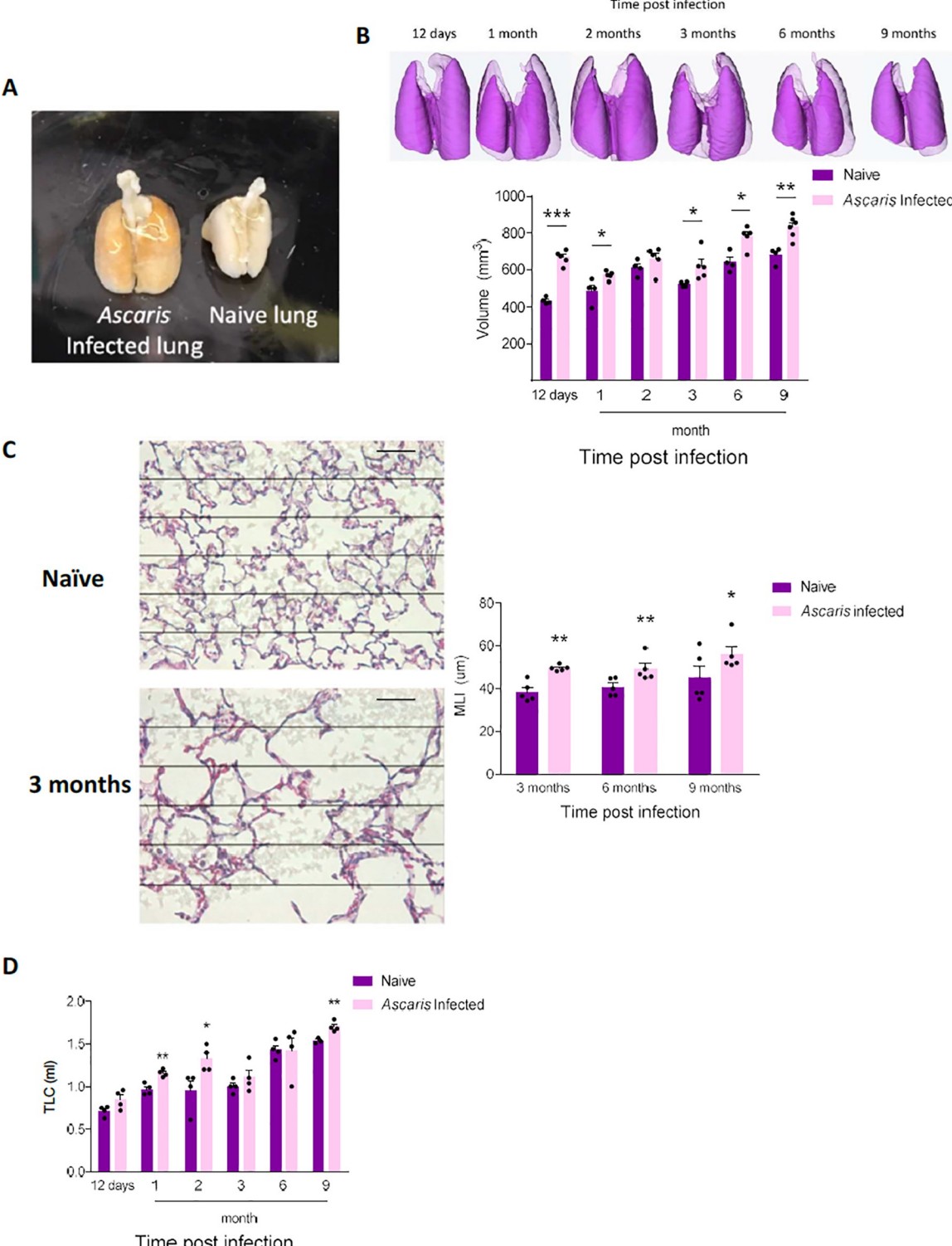

**Fig 2. *Ascaris suum* induces chronic lung disease.** Mice were challenged with *Ascaris suum* at standard time intervals as previously described. **(A)** Gross appearance of lungs from age- and sex-matched *Ascaris* challenged mice and PBS naïve mice at day 12 post-infection (p.i.) demonstrating visibly larger lungs in the *Ascaris* challenged mouse **(B)** Representative 3-D reconstruction and post-expiratory aerated lung volume from *Ascaris* challenged mice and PBS naïve mice was calculated using micro computed tomography (microCT) at standard intervals p.i. and **(C)** Mean linear intercept (MLI) calculation from lung histology of mice p.i. compared to PBS naïve control mice at 3 months, 6 months and 9 months p.i.. Sections of lungs were specifically selected based on larval transition

adjacent to bronchovascular bundles. **(D)** Lung compliance quantitated from mice at standard intervals p.i. measured as the total lung capacity (TLC) at a standard pressure (n≥5, mean±S.E.M, n.s.: not significant, *p<0.05, **p<0.01, ***p<0.001, using two tailed Student's t-test. Magnification: 400×, Grid and scale bar: 50μm. Data are representative of at least two independent experiments).

larval migration through the lungs induces chronic structural and functional changes that are similar to the effect of cigarette smoking leading to COPD.

### *Ascaris* larval migration induces chronic expression of MMP12 from alveolar and interstitial macrophages

Because *Ascaris* larval migration induced a chronic COPD-like phenotype (Fig 2), we hypothesized that expression of endogenous proteinases, especially elastases, linked to the development of emphysema[28,29] might also be upregulated. Thus, we assessed the production of matrix metalloproteinase (MMP)-9, MMP-12 and neutrophil elastase (ELANE) via qPCR. We discovered that MMP-12 mRNA level increased by more than 20-fold on average and remained increased even at 9-month p.i. (Fig 3A), whereas MMP-9 showed no difference (Fig 3B) and ELANE only displayed an acute response (Fig 3C). These data suggest that MMP-12, a potent mediator of smoking-related emphysema, may also be contributing to the chronic lung disease induced by *A.suum*.

Since macrophages are the primary source of most MMP-12[30], we next assessed alveolar and interstitial macrophages for MMP-12 production. Using flow cytometry, we gated on F4/80$^+$ CD64$^+$ CD11b$^-$ SigletF$^+$ alveolar macrophages and F4/80$^+$ CD64$^+$ CD11b$^-$ interstitial macrophages from whole lung of mice 3 months p.i. and quantified MMP-12 via intracellular staining. We discovered significantly more MMP12$^{high}$ alveolar macrophages (Fig 3D) and MMP12$^{high}$ interstitial macrophages (Fig 3E) from infected mice by percentile and absolute cell count, and higher overall MMP-12 level by MFI (Fig 3D and 3E). Together, we conclude that *A.suum* induces chronic upregulation of MMP-12 in the lungs contributed by alveolar and interstitial macrophages.

### *Acute Ascaris* infection induces chronic lung vascular damage and alveolar macrophage hemosiderosis

We next evaluated lung histopathology to understand more precisely the chronic effects of transient *Ascaris* infection and pulmonary larval migration. In addition to prominent emphysema, we noticed scattered inflammation surrounding broncho-vascular bundles and alveolar macrophages with atypical morphology as early as 1 month p.i. Under H&E staining, we identified macrophages with compact, dark-brown intracellular pigmentation similar in appearance to the anthracotic pigment of smokers' alveolar macrophages (Fig 4A). Whereas the anthracotic pigment of smokers is elemental carbon black[31], Prussian blue staining revealed that the intracellular pigment of *Ascaris*-infected mice was iron-positive (Fig 4B), and serial-block EM imaging confirmed that these cells were hemosiderin-laden macrophages evidenced by the presence of phagocytized erythrocytes (Fig 4C). Of note, these hemosiderin-laden macrophages were present in the lungs as long as 9-month p.i.

The hemosiderin-laden macrophages were also observed chronically in the BALF from *A. suum* infected mice (Fig 4D). Surprisingly, while the absolute macrophage cell count in the BALF decreased from 3 months to 9 months p.i. (Fig 1B), the absolute hemosiderin-laden macrophage cell count increased from 3 months to 9 months p.i. (Fig 4E). Furthermore, the percentage of hemosiderin-laden macrophages in the BALF increased over time, with the highest percentage noted at 9 months p.i. (Fig 4F).

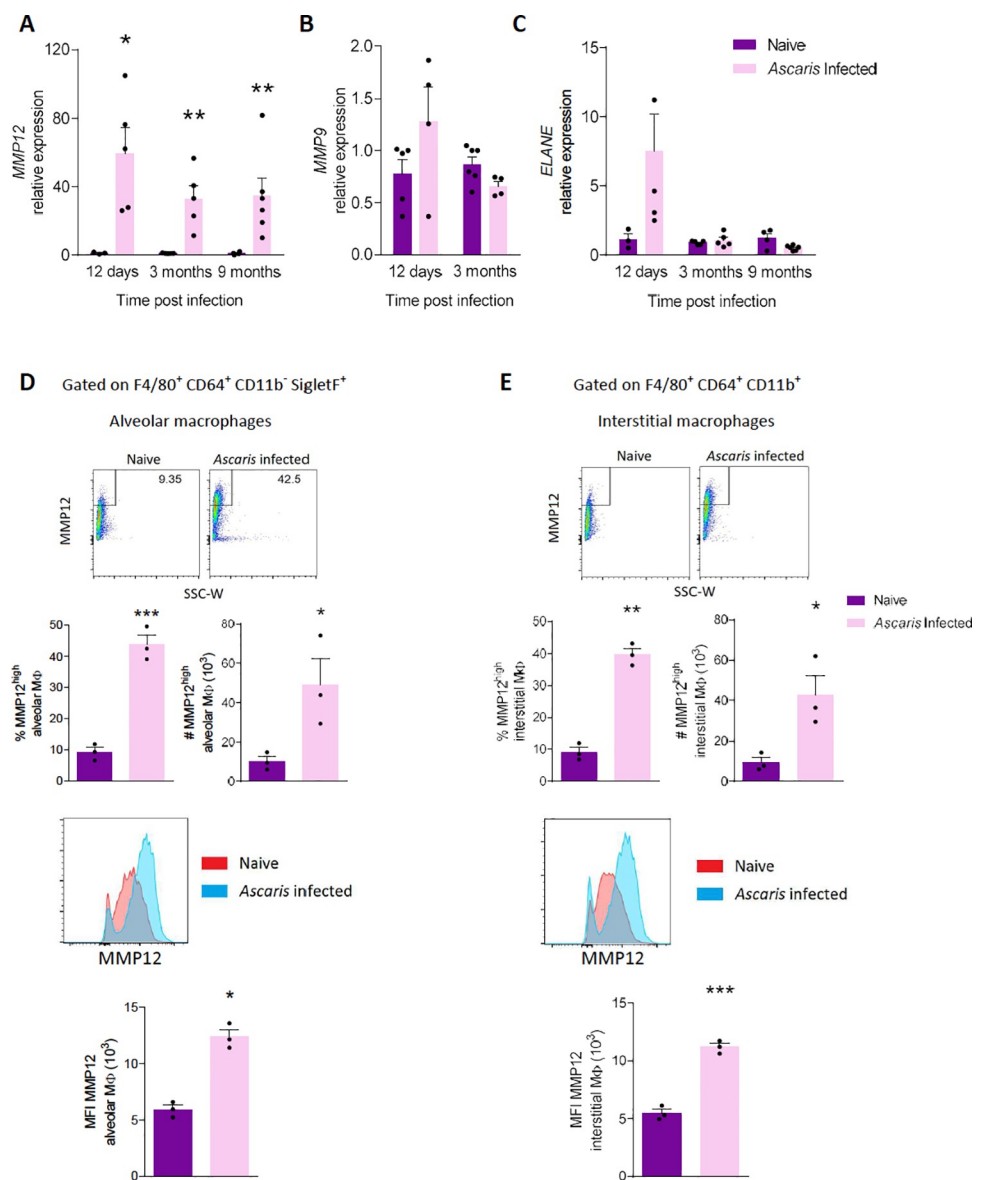

**Fig 3. Matrix Metalloprotease (MMP)-12 is chronically upregulated in alveolar and interstitial macrophages (MΦ) by _Ascaris_.** (A-C) mRNA level of **(A)** MMP12, **(B)** MMP9, and **(C)** neutrophil elastase (ELANE) from _Ascaris_ infected and naïve mice post infection. **(D-F)** Flow cytometry analysis on F4/80$^+$ CD64$^+$ CD11b$^-$ SigletF$^+$ Alveolar macrophages and F4/80$^+$ CD64$^+$ CD11b$^+$ interstitial macrophages **(D)** Representative flow plot of MMP12 on alveolar macrophages and the aggregate data of MMP12$^{high}$ alveolar macrophages expressed as percentages and absolute cell numbers and representative histograms and median fluorescence intensity (MFI) of MMP12 data are shown. **(E)** Representative flow plot of MMP12 on interstitial macrophages and the aggregate data of MMP12$^{high}$ interstitial macrophages expressed as percentages and absolute cell numbers and representative histograms and median fluorescence intensity (MFI) of MMP12 data are shown. (n≥3, mean ± S.E.M, n.s.: not significant, $^*p<0.05$, $^{**}p<0.01$, $^{***}p<0.001$, using two tailed Student's t-test. Data are representative of two independent experiments)

The presence of hemosiderin-laden macrophages in both lung tissue and BALF over time signaled the likelihood of chronic pulmonary vascular damage in _Ascaris_-infected mice, which was supported by the presence of erythrocytes within alveoli as long as 9 months p.i. (S3 Fig). As acute _Ascaris_ larval migration involves larvae traversing the vascular wall into the alveoli, we next determined vascular permeability after larval clearance. First, we quantified

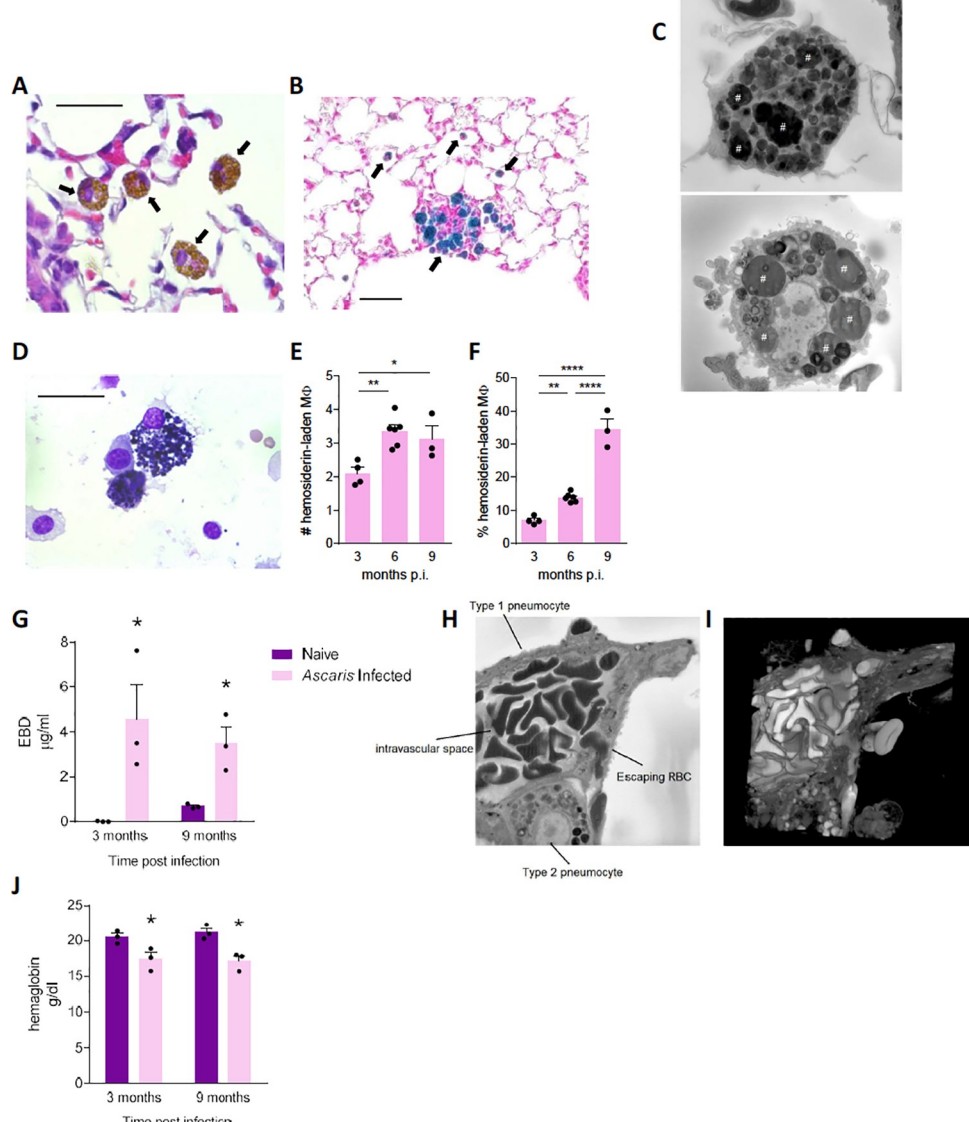

**Fig 4. *Ascaris* induced vascular disruption and chronically induced hemosiderin-laden alveolar macrophages (MΦ).** *Ascaris* infected and PBS-challenged, naive mice were euthanized and lungs were harvested for histopathologic analysis at the indicated time interval. **(A)** Haematoxylin and eosin (H&E), and **(B)** Prussian blue staining of 5 μm lung sections. Arrow represents hemosiderin-laden macrophage. **(C)** Serial block electron microscopy (EM) image of hemosiderin-laden macrophages from lung sections demonstrates engulfed erythrocytes noted by the pound sign. These hemosiderin-laden macrophages were identified at 3 months, 6 months and 9 months post-infection (p.i.). **(D)** H&E staining of bronchoalveolar lavage fluid (BALF) cells demonstrates hemosiderin-laden macrophage that were also identified at 3 months, 6 months and 9 months p.i.. Absolute cell count **(E)** and percentage **(F)** of hemosiderin-laden macrophages from BALF at 3, 6 and 9 months p.i.. demonstrating the % of hemosiderin-laden macrophages increases chronically **(G)** Pulmonary extravasation of Evan's blue dye (EBD) at 3 and 9 months p.i. to evaluate vascular integrity and leakage **(H)** EM image and **(I)** three-dimensional EM reconstruction of damaged pulmonary vascular structure demonstrating extravasation of the red blood cell (RBC) between a type-1 and type-2 pneumocyte moving from the endovasculature into the alveolar space due to edema. **(J)** Hemoglobin level from whole blood from infected mice at 3 and 9 months p.i. demonstrate chronic anemia in *Ascaris* infected mice. (Data are representative of at least two independent experiments)

accumulation of Evan's Blue Dye (EBD) in lungs after i.v. injection in mice to assess vascular permeability. Mice at 3 months and 9 months p.i. showed significantly higher retention of EBD in the lungs, indicating chronic vascular damage and permeability (Fig 4G). Using serial-

block EM, visualization of vascular damage was demonstrated by endothelial edema and erythrocyte extravasation from the endovasculature into the alveoli (Fig 4H and 4I). These results confirmed that *Ascaris* larval migration induces chronic pulmonary vascular damage leading to erythrocyte extravasation into the alveoli and subsequent uptake by macrophages.

In the setting of chronic lung vascular permeability and hemosiderin-laden macrophages, whole blood hemoglobin level in mice post *A.suum* infection was assessed to determine if the mice were anemic from chronic pulmonary blood loss. To our surprise, mice developed chronic anemia lasting up to 9 months p.i., marked by significantly decreased hemoglobin levels in whole blood (Fig 4J). Together these findings demonstrate that despite resolution of infection, chronic vascular damage continues to occur leading to the persistence of hemosiderin-laden macrophages and likely contributing to chronic anemia.

## Discussion

We have demonstrated that a single exposure to *A.suum* can lead to the development of chronic lung disease in two forms, pulmonary type-2 inflammatory pathology and emphysema-like disease, in a mouse model. Furthermore, we show that this chronic lung disease is associated with increased vascular permeability in the lungs and subsequent anemia months after infection resolution. Our results suggest, similar to other respiratory infections like tuberculosis and respiratory syncytial virus (RSV)[32–34], that ascariasis may be a risk factor for the development of non-communicable diseases like chronic asthma and/or emphysema as well as anemia in parasite-endemic regions of the world.

Previous studies have described the short-term relationship between *Ascaris* larval migration and allergic sensitization suggesting *Ascaris* larval migration may modulate allergic diseases like asthma or food allergy[12,35,36]. Our previous work specifically demonstrates that *Ascaris* larval migration causes type-2 inflammatory infiltrate and airway hyperreactivity consistent with short-term allergic airway disease in a mouse model[11]. Here, we have shown that a single standardized infectious dose of 2,500 *A.suum* eggs, the infectious dose previously determined to replicate acute human *Ascaris* pulmonary manifestations in a mouse model, can induce chronic pulmonary type-2 inflammatory pathology in mice months after infection resolution. Moreover, prolonged expression of classic type-2 cytokines and chemokines indicate that *A.suum* larvae are a potent pathogenic factor for the development of chronic pulmonary type-2 inflammatory pathology, similar to allergic airways disease. However, the molecular mechanism for the development of a type-2 immune response and airway hyperreactivity remains largely unknown. Potential mechanisms under investigation are focused on *Ascaris*-derived molecules that aid in larval migration through host tissue but also induce a polarized and sustained type-2 immune response.

In addition to inducing a chronic pulmonary type-2 inflammatory phenotype, *A. suum* also generated an emphysema-like phenotype in a mouse model. *A. suum* infection increased the size of the lungs and MLI while also increasing compliance chronically. The longevity of the emphysema phenotype suggests that endogenous factors play an important role, namely MMPs, in the phenotype pathogenesis. We verified the upregulation of MMP-12 expression from whole lung in *Ascaris*-challenged mice and identified high levels of MMP-12 from alveolar and interstitial macrophages. The direct contribution of MMP-12 in inducing emphysema has been previously described[37]. Thus, our results indicate that ascariasis can provoke the development of emphysema-like COPD in mice potentially through the persistent elevated expression of MMP-12 however further comprehensive evaluation is needed. The mechanism through which alveolar and interstitial macrophages maintain activation and high expression of MMP-12 remains largely unknown. It is possible that retained *Ascaris* products within the

lungs chronically activate alveolar and interstitial macrophages. While evaluation of lung histology did not reveal the presence of *Ascaris* larvae in the lungs after Day 12 p.i., proteomic analysis of infected lung tissue to identify retained *Ascaris* proteins over time is on-going. Alternative mechanisms including auto-antibody development against host tissue like the pulmonary endothelium and epithelium seen in certain COPD phenotypes[38,39], are also under investigation.

Recognition of non-communicable respiratory diseases such as asthma and COPD is increasing in LMIC[40]. While the highest prevalence of non-communicable respiratory diseases occurs in high-income countries (HIC), the greatest morbidity and mortality occur in LMIC[41–43]. Unlike in HIC, environmental factors such as indoor burning of biomass fuels, and not cigarette smoke, are more commonly linked to chronic lung disease in LMIC. Childhood factors that shape the growth and development of the lung, such as early respiratory infections, have also been found to increase susceptibility to asthma and COPD later in life [33,44]. Thus, within certain environments, the development of chronic lung disease may start early in life[44]. Ascariasis is a highly prevalent parasitic infection of children from LMIC that involves a highly immunogenic larval migratory phase through the lungs. The association of ascariasis and chronic severe asthma has been defined in clinical studies within LMIC[45,46]. While a clinical link between human ascariasis and COPD phenotypes has yet to be described, a mouse model of *Nippostrongylus braziliensis*, the rat hookworm routinely used as a surrogate of human hookworm in animal models, suggests similar emphysematous findings as described in this study, including the presence of hemosiderin-laden macrophages and high MMP-12 expression in mouse lungs[47]. Interestingly, using MMP-12 deficient mice, the Marsland *et al* study observed that the emphysematous histologic phenotype in *N. braziliensis* infected mice at 30 p.i. still developed. The authors concluded that MMP-12 may not play a role in the development of helminth-derived chronic lung disease despite detection of high levels. However, there was no quantitative analysis or long-term follow up to determine the chronic impact of MMP-12. Thus, the findings presented by Marsland *et al.* do not negate the role of MMP-12 but highlight the need for a more comprehensive MMP-12 evaluation. Furthermore, building on the Marsland *et al* hookworm model, this study provides the previously unreported physiologic and radiographic correlates of chronic lung disease development as a result of *Ascaris* larval migration in the lungs, a pathogen of significant human and zoonotic consequence. In contrast to the Marsland *et al* report, our study further demonstrates that *Ascaris* larval migration not only causes chronic structural lung disease but also results in vascular permeability in the lung leading to chronic anemia.

Long-term vascular damage after *Ascaris* larvae clearance, supported by the discovery of hemosiderin-laden macrophages and vascular extravasation of erythrocytes, was observed in this mouse model. Surprisingly, the pulmonary vascular permeability led to chronic anemia for months after resolution of the acute infection. Anemia is a common clinical finding in children infected with adult intestinal parasites as a result of malnutrition in the case of ascariasis and chronic blood loss due to hookworm[48,49]. However, as this mouse model is not competent for the development of adult intestinal ascariasis, our results indicate that chronic lung damage from larvae leads to anemia, an unexpected conclusion that is relevant to understanding the root causes of anemia in parasite-endemic regions.

Together, our study demonstrates that a single, transient exposure to *A. suum* leads to chronic pulmonary type-2 inflammatory pathology, COPD-like phenotype, pulmonary vascular permeability and anemia. These findings are likely dose-dependent and directly correlate with larval migration through the host lungs, suggesting *Ascaris* larvae are a critical risk factor for human chronic lung diseases and anemia in endemic regions. Our findings therefore support the development of interventions that prevent larval migration and *Ascaris* re-infection,

including preventative vaccines, to reduce *Ascaris*-induced morbidity that stunts the development of children globally. Furthermore, despite the immunomodulatory properties of adult intestinal helminths, the chronic lung disease observed as a result of both hookworm[47] and *Ascaris* further highlight the detrimental nature of helminth larval migration and should raise caution in clinical studies using these pathogens as therapeutic agents against non-communicable lung diseases like asthma[50,51]. However, targeted research on specific helminth-derived immunomodulatory molecules may provide future insight on the mechanisms of helminth-induced chronic lung disease as well as reveal potential therapeutic interventions with broad application against both helminths and non-communicable lung diseases.

## Methods

### Ethics statement

All experimental protocols were approved by the Institutional Animal Care and Use Committee of Baylor College of Medicine and followed federal guidelines.

### Mice

8 week-old female BALB/c mice (IMSR_JAX:000651) were purchased from Jackson Laboratories. Female mice were used to ensure consistency in infectious burden and of airway inflammation[52,53]. All mice were housed at the American Association for Accreditation of Laboratory Animal Care-accredited vivarium at Baylor College of Medicine under specific-pathogen-free conditions. Upon arrival, complete randomization of mice into longitudinal groups was performed.

### *Ascaris suum* experimental mouse model

Embryonated *Ascaris suum* eggs were kindly provided by Dr. Ricardo Fujiwara, Departamento de Parasitologia, Universidade Federal de Minas Gerais, Brazil. BALB/c mice were treated with a single inoculum of 2,500 fully embryonated *A. suum* eggs via oral gavage. The infectious dose of *Ascaris* has been standardized in the literature using egg titration in order to replicate human disease of eosinophilic pneumonitis and allergic airway disease in a mouse model[8,15]. Ascaris induced phenotypes in mice have been found to be dose-dependent[54] as a result the standardized infectious dose was used in this study. *Ascaris* infected and age-matched, PBS-challenged, naive mice were followed at intervals of 12 days post infection (p.i.), 1 month p.i., 2 months p.i., 3 months p.i., 6 months p.i. and 9 months p.i. Each time interval represents a separate, non-contiguous cohort of *Ascaris* infected and naive mice as physiologic procedures (airway hyperreactivity and compliance) were terminal. Each time interval included 5 mice per group. *Ascaris suum* lifecycle in a mouse model mimics an accelerated lifecycle in humans and has been previously described[11,15]. Briefly, *Ascaris* eggs hatch in the intestines, *Ascaris* larvae penetrate the intestinal epithelium and are carried to the lungs by the systemic circulatory system. *Ascaris* larvae have peak lung burden at day 8 p.i. The larvae leave the lungs, ascending the bronchotracheal tree on day 12 p.i. and are swallowed back into the intestines. Unlike human disease, the *Ascaris* mouse model is not competent to sustain chronic adult worms in the intestines. Thus, *Ascaris* larvae are subsequently excreted in the stool by day 14 p.i. The *Ascaris suum* experimental mouse model is only capable of evaluating the direct outcomes of *Ascaris* larval migration.

## Micro-computed tomography (microCT) analysis

*Ascaris* infected and age-matched naïve mice were anesthetized with isoflurane delivered by precision vaporizer by the Baylor College of Medicine Phenotyping Core at each designated time interval. Animals were under anesthesia and provided thermal support from a heating pad placed on the imaging cradle. Images were acquired using the Trifoil Explore CT120 imaging system and Console software following the "Live Lung-CT Continuous Rotation" imaging sequence (900 images, incremented 0.4° per image, X-Ray tube set at 90.0kV 40.0mA, detector set to 2x2 binning, with 16ms exposure time, gain set at 140 units and offset at 50 units). Three dimensional images were generated from the Trifoil Console applications' "Reconstruction" feature and a Ringfix Ring Correction Utility provided by Trifoil was used to preprocess images to reduce ring artifacts associated with CT imaging in the data. Reconstructed images were then exported as VFF and DICOM files for analysis. Raw data generated from the imager was analyzed using Amira 3.1.1 software for advanced image processing, quantification and 3-D rendering (Thermofisher scientific, Waltham MA). Values reported as "post-expiratory aerated lung volume" represent the period between the end of passive expiration and before the subsequent inhalation in anesthetized mice.

## Physiologic evaluation of chronic lung disease

Following microCT evaluation, induced airway hyperreactivity (AHR) was assessed at each designated time interval by bronchial constriction in response to acetylcholine (Ach) injected intravenously using whole body plethysmography. Briefly, mice were anesthetized with etomidate and intravenously injected with Ach via tail vein to assess AHR as quantified by measuring respiratory system resistance ($R_{RS}$). A separate age-matched cohort of mice of *Ascaris* infected and naïve mice were used for compliance evaluation at each designated time interval. Mice were euthanized with ketamine and xylazine and subsequently received a tracheostomy with a 20-gauge angiocatheter. Using a the pressure-volume syringe-pump system lungs were inflated to a pressure of 35 cm $H_2O$ followed by reversal of the pump until 0 cm $H_2O$. Data was collected using Powerlab software. As previously described, lung compliance which is the extent to which lungs expand, was estimated by total lung capacity (TLC) at a standard pressure[55].

## Collection and preparation of mouse tissues

Total bronchoalveolar lavage fluid (BALF) was collected by lavaging whole lung through the tracheostomy angiocatheter. Total cells from BALF were enumerated and differential cell counting was performed on modified Giemsa-stained cytospin preparations. Plasma was harvested by collecting whole blood into 10% 0.5M EDTA (Thermofisher scientific, Waltham MA) coated tubes. The right lung superior, middle, and inferior lobes were harvested and processed as follows. Lungs were cut into small pieces and incubated in digestion buffer (2mg/ml collagenase (#LS004177, Worthington), 0.04mg/ml DNAse (#10104159001, Sigma) 1, 20% FBS in HBSS) for 1 h at 37°C after which they were deaggregated by pressing through a 40 μM nylon mesh and centrifuged at 400 x g for 5 minutes at 4°C. Supernatants were discarded, and 1.5 mL of ACK (Thermofisher scientific, Waltham MA) was added and incubated for 3 min at room temperature for erythrocyte lysis. ACK was then neutralized with 7.5 mL of complete RPMI-1640 (Corning, NY), with 10% FBS and 1% Pen Strep, (Gibco, Waltham MA). The resulting leukocyte single cell solution was centrifuged and prepared for flow cytometry analysis, Luminex or ELISA.

## Cytokine and chemokine measurements

Single cell suspensions were cultured overnight at 37˚C and cell culture supernatants were collected and analyzed for cytokine and chemokine concentrations. Cytokines were first analyzed by standard ELISA after comparison to recombinant standard. IL-4, IL-5, IL-13 (DY404, DY405, DY413, R&D systems, Minneapolis, MN) were used as antibody pairs and capture signals were amplified using Streptavidin-horseradish peroxidase conjugate (51–9002813, BD Biosciences, San Jose, CA). The plate was further developed using TMB substrate solution (N301, Thermofisher scientific, Waltham MA) and detected at the absorbance wavelength of 450 nm. Alternatively, cytokine and chemokine analysis was done using Luminex MAGPIX equipment (Austin, Texas) and R&D Systems mouse magnetic Luminex assay (Catalog number LXSAMSM-19).

## Quantitative PCR

Total RNA was extracted using TRIzol from the right post-caval lobe (15596026, Thermofisher scientific, Waltham MA). Relative expression of mRNA for MUC5AC, MMP9, MMP12 and ELANE were detected by two-step, real-time quantitative reverse transcription-polymerase chain reaction (RT-PCR) with the ABI Perkin Elmer Prism 5700 Sequence Detection System (Applied Biosystems, Foster City, CA) using Taqman probe (Mm01276718, Mm00442991, Mm00500554, Mm00469310, Thermofisher scientific, Waltham MA). All data were normalized to eukaryotic 18s rRNA endogenous control (thermofisher scientific, Waltham, MA).

## Lung morphology

After removal of the right lung, the right main stem bronchi was tied and the left lung was inflated to 20 cm of water pressure, fixed in 10% neutral-buffered formalin solution, processed and embedded in paraffin. 5 $\mu$m sections were cut and slides were stained with H&E. Mean linear intercept (MLI), a measurement of airspace destruction and a metric used in experimental models of emphysema, was measured as previously described [25,56]. MLI was determined by selecting 10 fields from an H&E stained slide of the left lung by a blinded reviewer. Fields were randomly selected but needed to include a bronchovascular bundle within the view (S2 Fig). *Ascaris* larvae migration within the lungs is not uniform. However, larvae do pass through bronchovascular bundles which are the location of dense inflammatory infiltrate on day 12 p.i. as described previously[11,15]. Parallel lines were overlaid using Image J (40 µm apart) on serial lung images and intercepts were quantified by two blinded reviewers. MLI was calculated by multiplying the length and the number of lines per field (2455 µm), divided by the average number of intercepts[25]. The lung tissue was also evaluated for cellular infiltration and epithelium damage by H&E stain and for hemosiderin-laden macrophages by Prussian blue stain (catalog # 87001–921, VWR, Radnor, PA).

## Flow cytometry

Total lung cells isolated above were stained with Live/Dead Fix Blue (L34961, Thermofisher scientific, Waltham MA), F4/80, CD64, CD11b (123107, 139313, 101223, Biolegend, San Diego, CA), SigletF (740388, BD Biosciences, San Jose, CA), then permeabilized and fixed using Cytofix/Cytoperm Fixation/Permeablization Kit (554714, BD Biosciences, San Jose, CA), and stained for MMP12 (sc-390863 AF647, Santa Cruz, Dallas, TX) (S4 Fig).

## Serial block-face scanning electron microscopy

Mouse lungs were processed for serial block-face scanning electron microscopy (SBF-SEM) as described previously in detail[57]. Briefly, excised lungs were inflated through the trachea with fixative (0.1M sodium cacodylate buffer containing 2.5% glutaraldehyde), post-stained with heavy metals (Fe, $OsO_4$, uranyl acetate, lead) before dehydration through an acetone series and embedding in Embed 812 resin (Embed-812, Electron Microscopy Sciences, Hatsfield, PA) containing Ketjen Black (Ketjenblack EC600JD, Lion Specialty Chemicals Co., Tokyo). The resin-embedded blocks were sputter-coated with gold to reduce charging during block-face imaging. Tissue blocks were sectioned at 100–200 nm using a Gatan 3View2 system (Gatan, Pleasanton, CA) mounted to a Mira 3 scanning electron microscope (SEM, Tescan, Pittsburgh, PA) and serial images were recorded at a magnification ranging from 1,260–29,400x and pixel size from 4–15 nm.

## Lung permeability

Mice previously infected with *Ascaris* 3 or 9 months p.i. and age-matched, PBS-challenged, naïve mice received 60 mg/kg of Evan's blue dye (E2129, Sigma-Aldrich, St. Louis, MO) injected into the tail vein. The dye circulated for 1 hour and the mice were subsequently euthanized. Lungs were perfused with saline, extracted and incubated at 60˚C for 24 hours in formalin. The supernatant was measured at 620nm in a spectrophotometer and the total dye was calculated using a standard curve[58].

## Quantification of hemoglobin

Whole blood was collected from mice and anti-coagulated using 0.05M EDTA. 20μl of whole blood was added to 180μl of hemoglobin detector (#ab234046, Abcam, Cambridge MA) in 96 well plates and incubated at room temperature for 15 minutes. OD value was detected at 575 nm. Concentration was calculated through the standard curve.

## Statistical analysis

Data are presented as means ± standard errors of the means. Significant differences relative to PBS-challenged mice or appropriate controls are expressed by P values of <0.05, as measured two tailed Student's t-test or one-way ANOVA followed by Tukey's test for multiple comparison. Data normality was confirmed using the Shapiro-Wilk test.

## Supporting information

**S1 Fig. PAS staining on lung sections from mice post *Ascaris* infection.** *Ascaris* infected mice were euthanized and lungs were harvested for histopathology analysis at 9 months post infection. Periodic acid–Schiff. (PAS) staining of 5 μm lung sections. Magnification: 400x (TIFF)

**S2 Fig. Low magnification microscopy on lung sections from mice post *Ascaris* infection.** *Ascaris* infected mice were euthanized and lungs were harvested for histopathology analysis at (A) 12 days or (B) 9 months post infection. Haematoxylin and eosin (H&E) staining of 5 μm lung sections. Black box demonstrates area of (A) dense immune cell infiltrate and (B) hemosiderin-laden macrophages around bronchovascular bundles. Magnification: 20x and 200x. (TIFF)

**S3 Fig. *Ascaris* induces chronic pulmonary hemorrhage.** *Ascaris* infected mice were euthanized and lungs were harvested for histopathology analysis at 9 months post infection.

Haematoxylin and eosin (H&E) staining of 5 μm lung sections. Magnification: 400x and 100x
(TIFF)

**S4 Fig. Flow cytometry gating strategy for alveolar and interstitial macrophages.**
(TIFF)

## Acknowledgments

Conceptualization and review of the manuscript was provided by Dr. David Corry. Conceptualization of the project was provided by Dr. Peter Hotez. Technical support was provided by S. D. Halon for the serial block-face imaging. Gross image of *Ascaris* infected and naïve lung provided by Dr. Ana Maria Jaramillo.

## Author Contributions

**Conceptualization:** Yifan Wu, Ana Clara Gazzinelli-Guimaraes, Ricardo Fujiwara, Maria Elena Bottazzi, Jill E. Weatherhead.

**Data curation:** Yifan Wu, Evan Li, Morgan Knight, Grace Adeniyi-Ipadeola, Li-zhen Song, Alan R. Burns, Ana Clara Gazzinelli-Guimaraes, Jill E. Weatherhead.

**Formal analysis:** Yifan Wu, Evan Li, Morgan Knight, Grace Adeniyi-Ipadeola, Li-zhen Song, Alan R. Burns, Jill E. Weatherhead.

**Funding acquisition:** Jill E. Weatherhead.

**Investigation:** Yifan Wu, Evan Li, Morgan Knight, Grace Adeniyi-Ipadeola, Li-zhen Song, Ana Clara Gazzinelli-Guimaraes, Jill E. Weatherhead.

**Methodology:** Yifan Wu, Evan Li, Morgan Knight, Grace Adeniyi-Ipadeola, Li-zhen Song, Alan R. Burns, Ana Clara Gazzinelli-Guimaraes, Jill E. Weatherhead.

**Project administration:** Jill E. Weatherhead.

**Resources:** Jill E. Weatherhead.

**Supervision:** Li-zhen Song, Ricardo Fujiwara, Maria Elena Bottazzi, Jill E. Weatherhead.

**Writing – original draft:** Yifan Wu, Jill E. Weatherhead.

**Writing – review & editing:** Yifan Wu, Ricardo Fujiwara, Maria Elena Bottazzi, Jill E. Weatherhead.

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
