## [Decision Letter · Decision Letter 0]

19 Nov 2021

Dear Dr. Weatherhead,

Thank you very much for submitting your manuscript "Transient Ascaris suum larval migration induces intractable chronic pulmonary disease and anemia in mice" for consideration at PLOS Neglected Tropical Diseases. As with all papers reviewed by the journal, your manuscript was reviewed by members of the editorial board and by several independent reviewers. The reviewers appreciated the attention to an important topic. Based on the reviews, we are likely to accept this manuscript for publication, providing that you modify the manuscript according to the review recommendations. 

Please add the suggested citations to your discussion and then this manuscript will be good to go.

Sincerely,

Keke C Fairfax, PhD

Deputy Editor

Keke Fairfax

Deputy Editor

Please add the suggested citations to your discussion and then this manuscript will be good to go.

Reviewer's Responses to Questions

**Key Review Criteria Required for Acceptance?**

**Methods**

-Are the objectives of the study clearly articulated with a clear testable hypothesis stated?

-Is the study design appropriate to address the stated objectives?

-Is the population clearly described and appropriate for the hypothesis being tested?

-Is the sample size sufficient to ensure adequate power to address the hypothesis being tested?

-Were correct statistical analysis used to support conclusions?

-Are there concerns about ethical or regulatory requirements being met?

Reviewer #1: Methods are clear and experimental design and analysis are rigorous.

Reviewer #2: The objectives and hypothesis should be stated in the abstract. 

Further the objectives are not clearly described in the introduction: to determine chronic/long term the effects of a single Ascaris infection? 

The number of mice in each group has not been noted. What were the group sizes for the different treatments and time points, this could be clearly indicated in methods and/or figure legends?

**Results**

-Does the analysis presented match the analysis plan?

-Are the results clearly and completely presented?

-Are the figures (Tables, Images) of sufficient quality for clarity?

Reviewer #1: The analysis and figures are of high quality and together they thoroughly characterize the lung outcomes of infection.

Reviewer #2: results are clear

**Conclusions**

-Are the conclusions supported by the data presented?

-Are the limitations of analysis clearly described?

-Do the authors discuss how these data can be helpful to advance our understanding of the topic under study?

-Is public health relevance addressed?

Reviewer #1: Overall, the authors provide impactful results of public health significance for helminth parasite-induced morbidity.

Reviewer #2: Yes.The authors do a good job of highlighting how Ascarias may be a factor that drives long term lung pathology/disease.

**Editorial and Data Presentation Modifications?**

Reviewer #1: Please cite and discuss the findings in the context of these papers. This would provide important perspective of the results in the context of known allergic responses to helminth exposure

J Clin Invest. 2019;129(9):3686–3701. doi: 10.1172/JCI127963

PLoS Pathog. 2021 Mar 2;17(3):e1009337 doi:10.1371/journal.ppat.1009337.

J Allergy Clin Immunol. 2021 Apr;147(4):1393-1401.e7. doi: 10.1016/j.jaci.2020.12.650.

Reviewer #2: (No Response)

**Summary and General Comments**

Reviewer #1: Dr. Weatherhead and colleagues investigate the chronic consequences of Ascaris suum infection in the lung. They demonstrate long-term effects of a single infection, including inflammation, hemorrhage, anemia and lung structural changes with features similar to emphysema. Overall, this is an excellent and thorough study with high quality analysis and lung methodologies to investigate the lung outcomes of infection. These results have strong public health impact and potential translational significance since they employ a human parasite, which transiently infects the lungs of humans. There are some minor caveats to this study, notably the lack of discussion of some important recent papers that could frame the work better in the context of allergic inflammation - Please see section above.

Reviewer #2: This is a useful manuscript that shows that Ascaris can be modelled in the mouse to generate lung pathology. Currently no published human studies exist that demonstrate that exposure Ascaris exposure may cause a decline in lung function. Such studies would be very useful in supporting the relevance of this work. As such studies emerge the importance of the model presented here will be increased.

PLOS authors have the option to publish the peer review history of their article (what does this mean?). If published, this will include your full peer review and any attached files.

Reviewer #1: No

Reviewer #2: No

Figure Files:

Data Requirements:

Reproducibility:

References

---

## [Editor Report · Decision Letter 1]

3 Dec 2021

Dear Dr. Weatherhead,

We are pleased to inform you that your manuscript 'Transient Ascaris suum larval migration induces intractable chronic pulmonary disease and anemia in mice' has been provisionally accepted for publication in PLOS Neglected Tropical Diseases.

Best regards,

Keke C Fairfax, PhD

Deputy Editor

Keke Fairfax

Deputy Editor

---

## [Editor Report · Acceptance letter]

13 Dec 2021

Dear Dr. Weatherhead,

We are delighted to inform you that your manuscript, "Transient Ascaris suum larval migration induces intractable chronic pulmonary disease and anemia in mice," has been formally accepted for publication in PLOS Neglected Tropical Diseases.

Best regards,

Shaden Kamhawi

co-Editor-in-Chief

Paul Brindley

co-Editor-in-Chief
